# Validating the performance of organ dysfunction scores in children with infection: A cohort study

Shaojun Li[1,2◉], Tao Tan[3◉], Jing Li[2,4], Hongdong Li[1], Liang Zhou[1], Ke Bai[4], Li Xiao[5], Ximing Xu[5], Liping Tan[1,2]*

1 Emergency Department, Children's Hospital of Chongqing Medical University, National Clinical Research Center for Child Health and Disorders, Ministry of Education Key Laboratory of Child Development and Disorders, Intensive Care Unit, Children's Hospital of Chongqing Medical University, Chongqing, China, 2 Chongqing Key Laboratory of Infection and Immunity in Rare Pediatric Diseases, Chongqing, China, 3 Chongqing Health Statistics Information Center, Chongqing Municipal Health Commission, Chongqing, China, 4 Intensive Care Unit, Children's Hospital of Chongqing Medical University, Chongqing, China, 5 Big Data Engineering Center, Children's Hospital of Chongqing Medical University, Chongqing, China

◉ These authors contributed equally to this work.
* tanlp0825@hotmail.com

**Data Availability Statement:** "All relevant data are within the paper and its supporting information files(S1 Dataset.csv). "

## Abstract

### Purpose

We aimed to validate the performance of six available scoring models for predicting hospital mortality in children with suspected or confirmed infections.

### Methods

This single-center retrospective cohort study included pediatric patients admitted to the PICU for infection. The primary outcome was hospital mortality. The six scores included the age-adapted pSOFA score, SIRS score, PELOD2 score, Sepsis-2 score, qSOFA score, and PMODS.

### Results

Of the 5,356 children admitted to the PICU, 9.1% (488) died, and 25.1% (1,342) had basic disease with a mortality rate of 12.7% (171); 65.3% (3,499) of the patients were younger than 2 years, and 59.4% (3,183) were male. The discrimination abilities of the pSOFA and PELOD2 scores were superior to those of the other models. The calibration curves of the pSOFA and PELOD2 scores were consistent between the predictions and observations. Elevated lactate levels were a risk factor for mortality.

### Conclusion

The pSOFA and PELOD2 scores had superior predictive performance for mortality. Given the relative unavailability of items and clinical operability, the pSOFA score should be recommended as an optimal tool for acute organ dysfunction in pediatric sepsis patients.

**Funding:** The study was supported by Chongqing Science and Technology Bureau and Health Commission Joint Medical Project (2021MSXM025), awarded to SL, Chongqing Medical University Graduate Smart Medical Project (YJSZHYX202007), awarded to SL, and Program for Youth Innovation in Future Medicine, Chongqing Medical University, Chongqing 400014, China, awarded to JL. The funders had no role in study design, data collection and analysis, decision to publish, or preparation of the manuscript.

**Competing interests:** The authors declare that they have no conflicts of interest.

Elevated lactate levels are related to a greater risk of death from infection in children in the PICU.

## Introduction

Sepsis is a predominant condition associated with high morbidity and mortality and is a major health burden for children worldwide [1]. It is estimated that 25.2 million children suffer from sepsis annually, and approximately 3.3 million children with sepsis die every year [2]. Sepsis is a potentially fatal syndrome caused by acute organ abnormalities resulting from an abnormal host response to infection [3]. Since the Sepsis-3 criteria were published in 2016 [3], criteria for applying Sepsis-3 in children are lacking [4]. Sepsis-3 included the Sequential Organ Failure Assessment (SOFA) as a scoring system for organ dysfunction in adult sepsis patients based on a large cohort study that revealed that the SOFA score had superior discrimination of hospital mortality compared to other scores in patients with suspected or definite infection [5, 6]. However, the SOFA score developed and validated with adult data was inapplicable to children. Given that current data are insufficient to propose a particular scoring system [7–11], the 2020 consensus of the Surviving Sepsis Campaign (SSC) did not recommend a specific screening or diagnostic tool for septic shock and sepsis-associated organ dysfunction (SAOD) in children [12].

Recently, several scores assessing pediatric organ dysfunction, including the pediatric SOFA (pSOFA) score, systemic inflammatory response syndrome (SIRS) score, quick SOFA (qSOFA) score, Pediatric Logistic Organ Dysfunction-2 (PELOD-2) score, and Pediatric Multiple Organ Dysfunction Score (PMODS), were adapted and validated in a series of cohort studies [8, 9, 13–17]. Matics and colleagues, who utilized the age-adapted pSOFA score and validated the performance of the PELOD score, pSOFA score, PELOD2 score and PMODS, demonstrated that the pSOFA score had superior discrimination for hospital mortality in a large PICU cohort [8]. Schlapbach et al. revealed that the pSOFA score was a better predictor of hospital mortality than the qSOFA score or SIRS score in their cohort of children with definitive or suspected infection [9]. However, one of the cohorts included patients aged 21 years or younger, and the median age of the other cohort was 13 years. These cohorts were not representative of healthy children in the PICU and originated from developed countries [4, 18].

A retrospective cohort study was performed using a PICU electronic health record (EHR) database to validate and compare the performance of currently available age-adapted scores in children with suspected or confirmed infection, and we attempted to provide clinical evidence for selecting an optimal scoring system for pediatric guidelines for septic shock and SOAD. Given that the latest SSC guidelines do not recommend a specific organ dysfunction tool for children with sepsis, SIRS was included in this study. A total of six scores were applied to our cohort. The pSOFA or qSOFA score used mean arterial pressure (MAP) [9], the PELOD2 score was calculated excluding unavailable items [9, 10], the Sepsis-2 score was determined in accordance with the International Pediatric Sepsis Consensus Conference [7], and the SIRS score and PMODS were summarized on the basis of the original provenance [7, 11].

## Methods

### Study design, setting and participants

A single-center retrospective cohort study was performed, and patients aged four weeks to 18 years who were admitted to the PICU at the Children's Hospital of Chongqing Medical

University (CHCMU) in China with suspected or diagnosed infections from 2015 to 2021 were included. The CHUCMU is composed of two hospitals: Yuzhong Hospital, with a 32-bed PICU, and Liangjiang Hospital, with a 55-bed PICU. The data were obtained from the Sepsis Specialized Disease Database (SSDD) of EHRs in CHCMU, which included children in the PICU with definitive or suspected infection. The data captured from the SSDD were anonymous, and the authors could not identify individual participants during or after the data collection. The study was approved by the Ethics Committee of CHCMU. The protocol was registered in the Chinese Clinical Trial Registry (registration number: ChiCTR2100053198), an international clinical trial registry platform. Research data were accessed on December 31, 2021.

In the SSDD, suspected infection was defined when a patient was first given antibiotics combined with culture sampling within 24 hours or if culture sampling was obtained first and then antibiotics were given within 72 hours. Six scores, including the pSOFA score, age-specific SIRS score, PELOD2 score, Sepsis-2 score, age-adapted qSOFA score, and PMODS, were calculated based on the lowest values recorded during the first day in the PICU, after which we validated the performance of the six scores from the SSDD cohort (S1 and S2 Tables). Additionally, the alternate pSOFA (pSOFAal) and alternate qSOFA (qSOFAal) scores, which include systolic blood pressure (SBP) as a cardiovascular item, were compared with the pSOFA and qSOFA scores.

## Data collection

We defined hospital mortality as the primary outcome and hospital stay and length of stay in the PICU as secondary outcomes. Baseline variables, inflammatory markers and candidate score variables were extracted, and the lowest and highest records during the first day of PICU admission were collected from the SSDD. The SIRS and Sepsis-2 scores followed the 2005 consensus for pediatric sepsis [7]. Age-adapted cardiovascular subscores of the pSOFA or pSOFAal scores were calculated by the PELOD2 threshold of MAP or systolic blood pressure (SBP), and renal subscores were defined by the PELOD2 threshold of the serum creatinine level [9, 10]. Age-adapted qSOFA and qSOFAal scores were determined by using age-adjusted respiratory rates and the MAP or SBP according to the 2005 Pediatric Sepsis Consensus [5, 7, 9]. Given that pupillary dilatation data were unavailable, the PELOD2 score was calculated using the remaining items [10].

## Statistics

Continuous variables are presented as medians with interquartile ranges (IQRs) or means with standard deviations, and binary variables are expressed as numbers with *percentages*. A t test was used to analyze normally distributed data, the Wilcoxon rank sum test was applied to compare nonnormally distributed data, and Pearson's chi-squared test was used for categorical variables. An adjusted risk model was developed for hospital mortality based on the patients' baseline characteristics, which included age, sex and comorbidities, by a multivariable logistic regression model. Univariate or multivariate logistic regression models were applied to the six scores with or without the baseline model to estimate the associations between the models and the primary outcome. The discrimination ability of the models was evaluated by the area under the receiver operating characteristic curve (AUROC). Calibration plots were generated to determine the agreement between the predicted and observed outcomes, and Brier scores (BSs) were calculated to present the overall performance of all the models [19]. Decision curve analysis (DCA) and the clinical impact curve (CIC) were generated to assess the clinical net benefit of the models [20, 21]. Subgroup analyses were then performed for the primary outcome, and the patients were stratified by basic patient factors, including age group, sex,

comorbidities and lactate level. Missing data were accounted for using multiple imputation (S1 Fig). Data analyses were performed using R version 4.0.5 (R Foundation). A two-sided *p value* less than 0.05 was considered to indicate statistical significance.

## Results

### Study cohort

From 2015 to 2021, 5,609 pediatric admission records were collected, and when duplicated records were removed and encounters with missing outcomes were excluded, 5,356 children in the PICU with suspected infection were eligible for the final cohort. Table 1 shows that children younger than 2 years accounted for the majority of the sample. The proportion of males was slightly greater than that of females. The overall mortality rate was 9.1%, and the mortality rate was greater among children with comorbidities. The three most common comorbidities were respiratory diseases, hematological diseases and traumatic diseases (S2 Fig). The median PICU length of stay was 6 days, and the median hospital length of stay was 21 days.

**Table 1. Baseline characteristics of the children with infections in the cohort.**

| Characteristic | Total (n = 5356) | Survived (n = 4,868) | Died (n = 488) | p value |
|---|---|---|---|---|
| Age_group* | | | | <0.001 |
| <2 yr | 3499 (65.3%) | 3231 (66.4%) | 268 (54.9%) | |
| >12_to_18_yr | 299 (5.6%) | 254 (5.2%) | 45 (9.2%) | |
| >5_to_12_yr | 787 (14.7%) | 694 (14.3%) | 93 (19.1%) | |
| 2_to_5yr | 771 (14.4%) | 689 (14.2%) | 82 (16.8%) | |
| Sex* | | | | 0.226 |
| Male | 3183 (59.4%) | 2906 (59.7%) | 277 (56.8%) | |
| Female | 2173 (40.6%) | 1940 (40.3%) | 211 (43.2%) | |
| WBC ($10^9$/L)# | 11.96 (8.09,17.30) | 11.92 (8.21, 17.11) | 12.79 (5.83, 20.21) | 0.84 |
| CRP (mg/L)# | 16.00 (5.00, 42.00) | 15.00 (5.00, 40.00) | 30.00 (5.00, 67.00) | <0.001 |
| PCT (ng/L)# | 0.98 (0.17, 7.47) | 0.85 (0.16, 5.94) | 3.85 (0.56, 26.91) | <0.001 |
| Lac (mmol/L)# | 1.80 (1.20, 3.00) | 1.70 (1.10, 2.70) | 5.30 (2.60, 9.20) | <0.001 |
| pSOFA* | 4.00 (3.00, 6.00) | 4.00 (3.00, 6.00) | 7.00 (5.00, 10.00) | <0.001 |
| SIRS* | 2.00 (1.00, 3.00) | 2.00 (1.00, 3.00) | 3.00 (2.00, 4.00) | <0.001 |
| Sepsis-2* | 2.00 (1.00, 2.00) | 2.00 (1.00, 2.00) | 3.00 (2.00, 4.00) | <0.001 |
| qSOFA* | 1.00 (1.00, 2.00) | 1.00 (1.00, 2.00) | 2.00 (1.00, 2.00) | <0.001 |
| PMODS* | 5.00 (4.00, 7.00) | 5.00 (4.00, 7.00) | 8.00 (6.00, 10.00) | <0.001 |
| PELOD2* | 6.00 (5.00, 7.00) | 6.00 (5.00, 7.00) | 8.00 (7.00, 9.00) | <0.001 |
| Comorbidities* | 1342 (25.1%) | 1171 (24.1%) | 171 (35.0%) | <0.001 |
| MV* | 4690 (87.5%) | 4202 (86.3%) | 488 (100%) | <0.001 |
| Hospital length of stay (day)# | 21.00 (12.00, 32.00) | 22.00 (13.00, 33.00) | 9.00 (3.00, 19.00) | <0.001 |
| PICU length of stay (day)# | 6.00 (3.00, 12.00) | 6.00 (3.00, 12.00) | 3.00 (1.00, 10.00) | <0.001 |
| ECMO* | 22 (0.4) | 13 (0.3) | 9 (1.8) | <0.001 |
| VP* | 958 (17.9) | 656 (13.5) | 302 (61.9) | <0.001 |
| RRT* | 266 (5.0) | 179 (3.7) | 87 (17.8) | <0.001 |

**Note:**

* n (%)

# Median (IQR); Abbreviations: WBC, white blood cell; CRP, C-reactive protein; PCT, procalcitonin; Lac, lactate; pSOFA, Pediatric Sequential Organ Failure Assessment; SIRS, systemic inflammatory response syndrome; qSOFA, quick SOFA; PMODS, Pediatric Multiple Organ Dysfunction Score; PELOD2, Pediatric Logistic Organ Dysfunction-2; MV, mechanical ventilation; ECMO, extracorporeal membrane oxygenation; VP, vasopressor; RRT, renal replacement therapy.

## Models and primary outcomes

Most patients had a pSOFA score, PELOD2 score, PMODS, SIRS score or Sepsis-2 score of 2 or higher, whereas half of the children had a qSOFA score of 2 or lower (S3 Fig). The pSOFAal and qSOFAal scores were similar to the pSOFA and qSOFA scores (S3 Table). Compared with the surviving patients, the patients who died had significantly greater pSOFA scores, PELOD2 scores, SIRS scores, Sepsis-2 scores, qSOFA, scores and PMODSs (Table 1). For the pSOFA score, PELOD2 score, PMODS, qSOFA score and Sepsis-2 score, patients with scores of 2 or more points had more than a 4-fold increased risk of mortality than patients with a 2-fold change in the SIRS score (Figs 1 and S4).

In the cohort, six scores with or without the adjusted baseline model were associated with a similar odds ratio (OR) and 95% confidence interval (95% CI) for in-hospital mortality [pSOFA score: 1.59 (1.53–1.65), 1.57 (1.51–1.63); SIRS: 1.46 (1.34–1.6), 1.39 (1.26–1.54); PELOD2 score: 2.1 (1.97–2.24), 2.07 (1.94–2.21); Sepsis-2 score: 2.71 (CI 2.46–2.98), 2.65 (2.41–2.92); qSOFA score: 4.28 (3.65–5.01), 4.25 (3.60–5.02); PMODS: 1.44 (1.39–1.5), 1.44 (1.38–1.49); un or adjusted model] (Fig 2).

## Performance of the models

Among the six scores, the pSOFA (AUROC 0.78, 95% CI 0.76–0.81) and PELOD2 (AUROC 0.79, 95% CI 0.77–0.81) scores had the highest discrimination, the Sepsis-2 score (AUROC 0.75, 95% CI 0.72–0.77) and PMODS (AUROC 0.74, 95% CI 0.72–0.77) had moderate discrimination, and the SIRS (AUROC 0.62, 95% CI 0.59–0.64) and qSOFA (AUROC 0.71, 95% CI, 0.68–0.73) scores had relatively poor discrimination. For the primary outcome, the AUC of each score adjusted for in the baseline model was similar to that of each score separately (Fig 3).

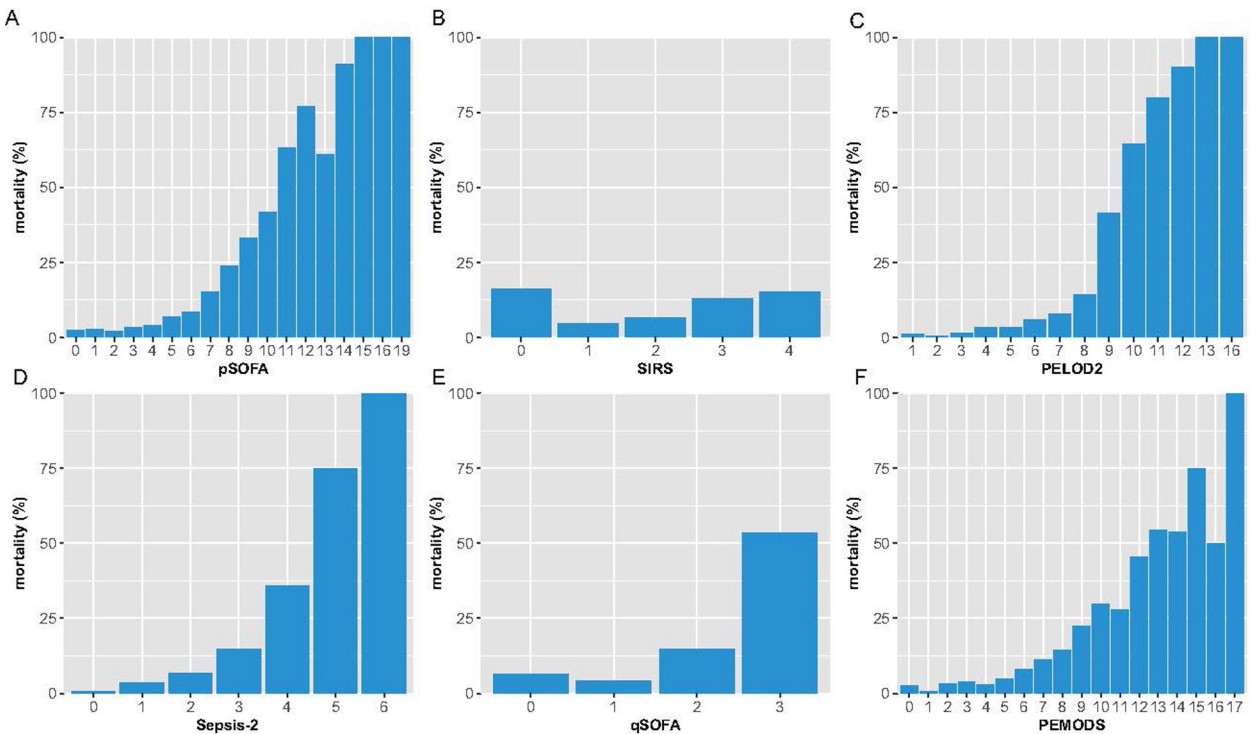

**Fig 1. Bar chart of the mortality distribution for each model score.** A pSOFA score, B SIRS score, C PELOD2 score, D Sepsis-2 score, E qSOFA score, F PMODS.

## A

| Model unadjusted | OR(95% CI) | Forest plot | Pvalue |
|---|---|---|---|
| pSOFA | 1.59(1.53−1.65) | | <0.001 |
| SIRS | 1.46(1.34−1.6) | | <0.001 |
| PELOD2 | 2.1(1.97−2.24) | | <0.001 |
| Sepsis-2 | 2.71(2.46−2.98) | | <0.001 |
| qSOFA | 4.28(3.65−5.01) | | <0.001 |
| PMODS | 1.44(1.39−1.5) | | <0.001 |

## B

| Model adjusted | OR(95% CI) | Forest plot | Pvalue |
|---|---|---|---|
| pSOFA | 1.57(1.51−1.63) | | <0.001 |
| SIRS | 1.39(1.26−1.54) | | <0.001 |
| PELOD2 | 2.07(1.94−2.21) | | <0.001 |
| Sepsis-2 | 2.65(2.41−2.92) | | <0.001 |
| qSOFA | 4.25(3.60−5.02) | | <0.001 |
| PMODS | 1.44(1.38−1.49) | | <0.001 |

**Fig 2. Unadjusted and adjusted scores according to the baseline model regressed for hospital mortality among children with suspected or confirmed infection.** Note: OR, odds ratio; 95% CI, 95% confidence interval.

Among the unadjusted scores, the calibration curves of the pSOFA and PELOD2 models in isolation presented superior consistency between the prediction and observation results, with superior overall performance, and the remaining four scores showed poor agreement with relatively poor overall performance (Fig 4). All the models adjusted by the baseline risk model for hospital mortality had nearly the same calibration and overall performance as the unadjusted models.

The DCA plots of the six scores indicated that the net benefits of the pSOFA and PELOD2 scores were greater than those of the other four scores, indicating that the former were optimal, while the latter were inferior (S7 Fig). As shown by the CIC, the pSOFA and PELOD2

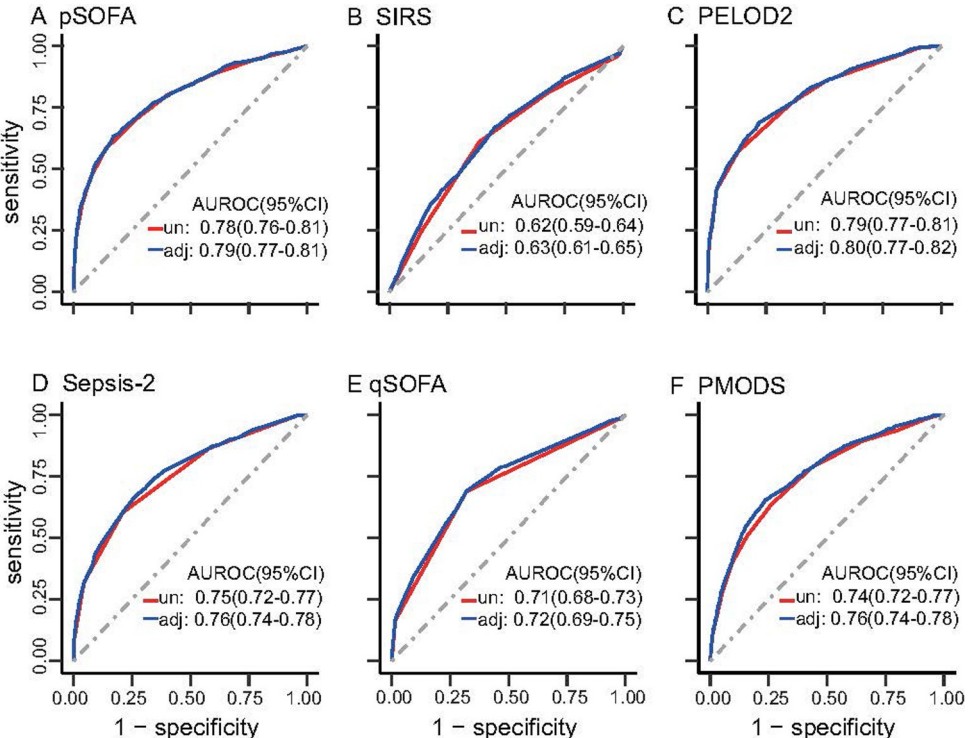

**Fig 3. Comparison of the AUROCs of unadjusted and adjusted scores for hospital mortality.** AUROCs are reported for A. pSOFA, B. SIRS, C. PELOD2, D. Sepsis-2, E. qSOFA, and F. PMODS. Note: AUROC, area under the receiver operating characteristic curve; un, unadjusted; adj, adjusted.

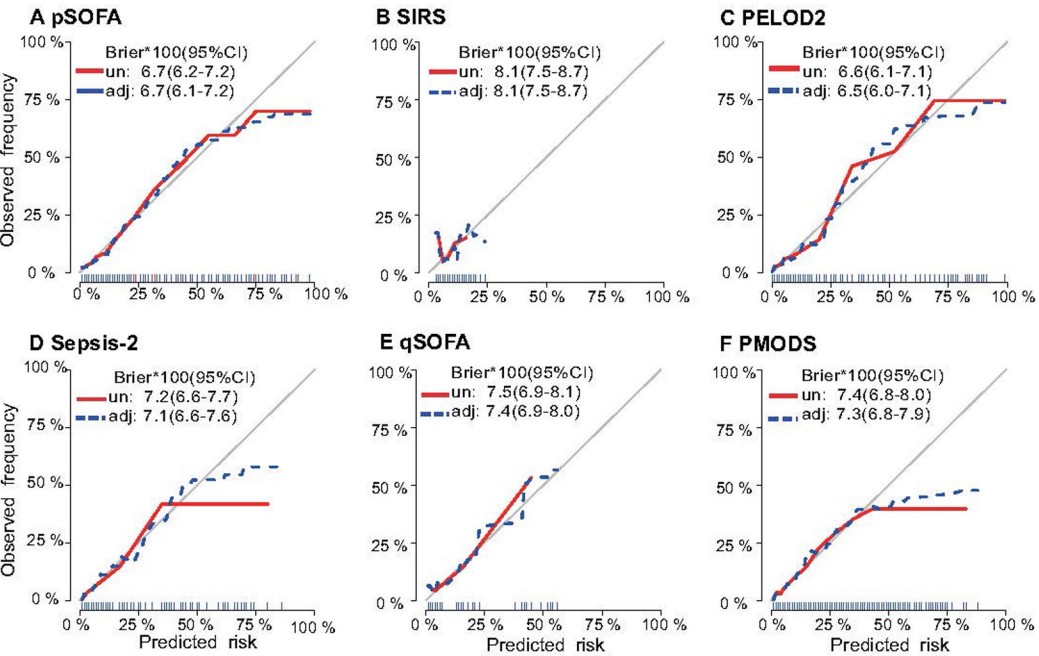

**Fig 4. Calibration plots of scores unadjusted and adjusted for in the baseline model for hospital mortality among all children in the PICU cohort.**

scores were superior in terms of overall net benefit, with relatively wide threshold probabilities in comparison with the other scores, which indicated that the former possessed significant clinical applicability.

Regarding discrimination and calibration, there were no differences between the pSOFAal and pSOFA scores or between the qSOFAal and qSOFA scores either with or without the baseline model (S3 Table).

## Subgroup analyses

Subgroup analyses were performed by age (groups aged $\geq$ 2 years or < 2 years), sex (male or female) and comorbidities (presence or absence of comorbidities) as the primary outcomes. The primary outcome exhibited similar changes between the patients in subgroup aged 2 years or older and those in subgroup aged younger than 2 years (S4 Fig). The data showed that the mortality of infected children increased with increasing lactate levels, and the mortality rate reached more than 30% when the lactate level was greater than 5 mmol/L (S5 Fig). In addition, lactate levels were positively correlated with pSOFA scores (S6 Fig). The levels of the inflammatory markers PCT and CRP were significantly greater in the nonsurviving group than in the surviving group (Table 1). Additionally, there was a relatively weak association between PCT or CRP levels and the six scoring systems (adjusted $R^2$ ranging from 0.02 to 0.17) (Figs S9 and S10).

## Discussion

In this cohort of PICU patients with infection in Southwest China, we validated six available scores to identify organ dysfunction in children with sepsis. The pSOFA score and PELOD2 score showed superior discrimination for hospital mortality compared with the other four models. These two scores also revealed good calibration between the prediction and observation and better clinical net benefit prediction than did the other scores for hospital mortality. In most PICUs, the MAP is calculated simultaneously by an electrocardiograph monitor when measuring SBP; therefore, for the pSOFA score, the MAP and SBP may be alternative cardiovascular indicators. Given the clinical applicability of these models, the pSOFAal score was calculated, and the SBP was used to replace the MAP-cardiovascular item of the pSOFA score, as in the qSOFAal score. Our data revealed that there was no difference in the use of MAP or SBP for either the pSOFA or qSOFA score. Compared with the PELOD2 score, the pSOFA score is more likely to be applied in clinical practice because it has fewer items. Therefore, the age-adapted pSOFA score should be considered when determining sepsis-associated organ dysfunction scores in children.

Since the sepsis-3 definition and criteria were established for adults in 2016, quite a few efforts have been made to apply these criteria in pediatric patients [8, 9, 15, 16, 18, 22–24]. A range of small- to medium-sample cohort studies assessed the prognosis of children with different clinical conditions in the PICU using the SIRS, pSOFA, and qSOFA scores, and the data were not sufficient to provide an optimal tool for identifying organ abnormalities in patients with sepsis. For children outside the PICU, high-quality studies evaluating sepsis-associated organ dysfunction are lacking, although several cohort studies have developed predictive models using machine learning algorithms to predict the occurrence of sepsis [23–25]. In recent years, an Australian and Zelanian PICU cohort included 2,715 children, and that study validated the prognostic accuracy of the pSOFA score, SIRS score, severe sepsis score, PELOD2 score and qSOFA score, revealing that the pSOFA score was better than the SIRS, Sepsis-2 and qSOFA scores [9]. Another American cohort in the PICU included 6,303 children aged younger than 21 years and used the pSOFA score, in which the MAP was separated into seven subgroups according to age; the pSOFA score exhibited superior performance in comparison with

other organ dysfunction scoring systems, such as the PELOD score, PELOD2 score and PMODS [8]. These two cohort studies and our study recruited participants admitted to the PICU and presented similar accuracy among recently available organ dysfunction models. With our data, additional calibration and clinical applicability were performed, which further validated the precision and clinical benefits of the scores, and the results showed that the pSOFA score had superior predictive performance for poor prognosis in the PICU. In 2020, the updated international guidelines for SSC in children defined septic shock and SAOD and suggested systematic screening to identify SAOD or septic shock but did not provide special criteria for organ dysfunction [12]. Recently, the Pediatric Organ Dysfunction Information Update Mandate, which derived an evidence-based organ dysfunction list for ten organs from published literature reviews, provided the latest roadmap for pediatricians to conduct and validate single- or multiple-organ dysfunction in children with sepsis [26]. Therefore, this study is an important supplement to clinical evidence [12].

The blood lactate value is used to evaluate tissue perfusion in sepsis or septic shock patients, and in most settings, lactate can be measured rapidly [12, 27]. Lactate levels are recommended as a marker of tissue hypoxia in children according to sepsis guidelines, but unlike adult guidelines, the pediatric guidelines do not suggest a certain lactate level threshold as an indication of metabolic abnormalities in septic shock patients [1, 12, 28, 29]. Our study revealed that lactate levels are an important predictor of hospital mortality in children with infections in the PICU. A series of observational studies indicated that elevated lactate levels were correlated with poor outcomes in children with septic shock [16, 28–30]. In a sepsis cohort from the PICU, lactate levels were independently associated with hospital mortality, and patients whose lactate levels were 2 mmol/L or greater had a twofold increased risk of in-hospital death [16]. Future studies should be performed to define the optimal threshold for hyperlactatemia in children.

Our cohort had relatively different demographics compared to those of other larger cohorts. In our Asian cohort, approximately two-thirds of the children were aged less than 2 years. In previous cohorts, white and black patients were included in the American cohort, and brown and white patients were included in the Australian cohort, accounting for nearly 30% of the patients younger than 2 years [8, 9]. Hence, these three large PICU cohorts possessed adequate population representativeness. In these studies, the varied distributions of hospital mortality might be attributed to the severity of the patients' conditions [8, 9]. In our cohort, 9.1% of the children died, and there was a greater proportion (12.7%) of patients with comorbidities. The mortality rate was 2.6% in the American cohort and 5.8% in the Australian cohort. Additionally, sensitivity analyses were performed with a baseline risk model to increase the robustness of the statistical results, and available organ dysfunction model-adjusted or unadjusted baseline models presented with similar ORs and discrimination ability for the primary outcome.

This study has several limitations. First, given the retrospective cohort of children as young as seven years based on EHRs and in which follow-up for discharged children was very difficult, we evaluated only hospital mortality as the primary clinical endpoint. A proportion of participants may have died in the first weeks after discharge. Second, our cohort included only patients in the PICU and did not include hospitalized children outside the PICU because of the large amount of missing data for non-PICU patients; therefore, we could not validate the models' performance for this population. Third, a small portion of patients in this PICU cohort had missing laboratory data, which may have affected the accuracy of the calculations. Fourth, the PELOD2 score was calculated without including pupillary dilation because it could not be extracted from the retrospective database. Fifth, we captured the lowest scores using the models on the first day of the PICU stay, so the results of this study were restricted to those after the first day of admission to the PICU. Researchers previously revealed that small-scale

clinical features in the first hour allowed the derivation of robust severity assessments for sepsis-associated mortality in children.

In conclusion, we validated six recently available organ dysfunction models for in-hospital mortality in a PICU infection cohort. Compared to the Sepsis-2 score, qSOFA score, SIRS score and PMODS, the pSOFA and PELOD2 score had superior accuracy for predicting hospital death, good precision between predictions and observations and better clinical utility in hospital mortality prediction. Given the relative unavailability of items and the clinical operability of the PELOD2 score, it cannot be considered an optimal tool, although it has a similar predictive performance for mortality as the pSOFA score. Therefore, the pSOFA score should be recommended as an optimal tool for predicting acute organ dysfunction in pediatric patients with sepsis. Elevated lactate levels are related to a greater risk of death from infection in children in the PICU. A combination of both scores should be used to identify organ dysfunction earlier in sepsis patients.

## Supporting information

**S1 Fig. Distribution of missing data on candidate features for children with suspected or confirmed infections admitted to the PICU in the cohort (n = 5356).**
(PDF)

**S2 Fig. Distribution of comorbidities among all children with suspected or confirmed infections admitted to the PICU in the cohort.** Note: The top three underlying diseases were respiratory diseases, hematological diseases and traumatic diseases. Abbreviations: resp, respiratory diseases; homo, hematological diseases; trau, traumatic diseases; heart, heart diseases; nutr, nutritional diseases; immu, autoimmune diseases; neur, neurological diseases; tumo, tumors; inhe, inherited metabolic disorders; CKD, chronic kidney disease.
(PDF)

**S3 Fig. Distribution of the candidate models for all children with suspected or confirmed infections admitted to the PICU in the cohort.** Note: The score distributions of the numbers of encounters are shown for A. pSOFA, B. pSOFAal, C. SIRS, D. PELOD2, E. Sepsis-2, F. qSOFA, G. qSOFAal, and H. PMODS.
(PDF)

**S4 Fig. Subgroup analyses by age group, sex and comorbidities as the primary outcome in children with 2 or more model points versus those with fewer than 2 model points.** Note: Mortality is shown for A. the group aged less than 2 years, B. the group aged 2 years or older, C. the male group, D. the female group, E. the group with comorbidities, F. the group without comorbidities, and G. the total cohort.
(PDF)

**S5 Fig. Bar chart of mortality changes associated with different lactate concentrations.**
(PDF)

**S6 Fig. Scatter plot of the pSOFA score vs. lactate level for the survival group and death group according to the linear regression line.**
(PDF)

**S7 Fig. DCAs for the candidate models.** A. pSOFA, B. SIRS, C. PELOD2, D. Sepsis-2, E. qSOFA, F. PMODS. Note: DCAs show that the pSOFA and PELOD2 scores have superior net benefits compared to the other five scores.
(PDF)

**S8 Fig. CICs for the candidate models.** A. pSOFA, B. SIRS, C. PELOD2, D. Sepsis-2, E. qSOFA, F. PMODS. Note: The pSOFA and PELOD2 scores are clinically applicable for in-hospital mortality prediction. Abbreviations: CIC, clinical impact curve.
(PDF)

**S9 Fig. Scatter plot of the six scores vs. PCT level.** A. pSOFA vs. PCT, B. SIRS vs. PCT, C. PELOD2 vs. PCT, D. Sepsis-2 vs. PCT, E. qSOFA vs. PCT, F. PMODS vs. PCT.
(PDF)

**S10 Fig. Scatter plot of the six scores vs. CRP level.** A. pSOFA vs. CRP, B. SIRS vs. CRP, C. PELOD2 vs. CRP, D. Sepsis-2 vs. CRP, E. qSOFA vs. CRP, F. PMODS vs. CRP.
(PDF)

**S1 Table. Scoring rules for the candidate models.**
(DOCX)

**S2 Table. Variables of the scoring models and the number of missing values for each variable.**
(DOCX)

**S3 Table. Performance of the pSOFAal vs. pSOFA scores and qSOFAal vs. qSOFA scores.**
(DOCX)

**S4 Table. Subgroup analyses by age group, sex and comorbidities for endpoint events of the primary outcome in children with 2 or more model points versus those with fewer than 2 model points.**
(DOCX)

**S1 Dataset. Original dataset.**
(CSV)

## Acknowledgments

We thank the members of the Big Data Center for Children's Medical Care, Children's Hospital of Chongqing Medical University and Shanghai Synyi Medical Technology Co., Ltd., for providing the database platform.

## Author Contributions

**Conceptualization:** Shaojun Li, Liping Tan.

**Data curation:** Tao Tan, Ke Bai, Li Xiao, Ximing Xu.

**Formal analysis:** Shaojun Li, Ke Bai, Ximing Xu.

**Funding acquisition:** Shaojun Li.

**Investigation:** Li Xiao.

**Methodology:** Shaojun Li, Liping Tan.

**Project administration:** Jing Li.

**Resources:** Tao Tan, Hongdong Li, Liang Zhou, Ximing Xu.

**Software:** Shaojun Li, Hongdong Li, Liang Zhou.

**Supervision:** Hongdong Li.

**Validation:** Liang Zhou.

**Visualization:** Jing Li.

**Writing – original draft:** Shaojun Li.

**Writing – review & editing:** Liping Tan.

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
