## [Decision Letter · Decision Letter 0]

4 Jan 2024

PONE-D-23-35521A cohort study validated the performance of organ dysfunction scores in children with infectionPLOS ONE

Dear Dr. Tan,

Thank you for submitting your manuscript to PLOS ONE. After careful consideration, we feel that it has merit but does not fully meet PLOS ONE’s publication criteria as it currently stands. Therefore, we invite you to submit a revised version of the manuscript that addresses the points raised during the review process.

We look forward to receiving your revised manuscript.

Kind regards,

Dong Wook Jekarl

Academic Editor

PLOS ONE

“The study was supported by Chongqing Science and Technology Bureau and Health Commission Joint Medical Project (2021MSXM025), Chongqing Medical University Graduate Smart Medical Project (YJSZHYX202007), and Program for Youth Innovation in Future Medicine, Chongqing Medical University, Chongqing 400014, China.”

Reviewers' comments:

Reviewer's Responses to Questions

**Comments to the Author**

1. Is the manuscript technically sound, and do the data support the conclusions?

Reviewer #1: Partly

Reviewer #2: Yes

2. Has the statistical analysis been performed appropriately and rigorously? 

Reviewer #1: Yes

Reviewer #2: Yes

3. Have the authors made all data underlying the findings in their manuscript fully available?

Reviewer #1: No

Reviewer #2: Yes

4. Is the manuscript presented in an intelligible fashion and written in standard English?

Reviewer #1: No

Reviewer #2: No

5. Review Comments to the Author

Reviewer #1: The authors present a retrospective cohort study on the validity of organ dysfunction scores in children with suspected infection admitted to the pediatric intensive care unit. The authors should be commended for their effort to analyse this large study population but I have several concerns.

Major concerns:

- Recently the PODIUM organ dysfunction scoring system has been proposed (PMID: 34970673). I highly recommend the authors assess the possibility to include this scoring system into their analysis to make their analysis more relevant. At the least this new organ dysfunction scoring system should be discussed in their manuscript to keep the discussion of the relevant literature accurate.

- The authors state they compare 8 organ dysfunction scores, but alternate SOFA, and alternate qSOFA only swap MAP for SBP. The results of this comparison could be presented as supplementary analysis and IMHO do not qualify for inclusion as separate scores. (the results of the analysis could probably be abbreviated that there was no difference in the use of MAP or SBP).

- The authors should also clarify why they included SIRS, which is not an organ dysfunction scoring system and has been removed from the definition of adult sepsis 7 years ago (obviously pediatric definitions are not yet published but will adopt this as well). This is relevant as the authors presumably report on a very sick population (>85% with mechanical ventilation) but only 70% had a SIRS score >=2, which begs the question how relevant this measure could be in their population?

- Discussion: your interpretation of the accuracy, calibration and net benefit of the scores seems a bit euphoric. They can hardly be called excellent and outstanding.

- Judgement of usefulness of PELOD-2 based on the availability of pupillary dilatation in your database is moot. This can be easily measured and is a failure of the design of the database for the analysis presented here

- please expand on the limitations section (e.g. children did not have proven infection, data validity for other settings, etc.)

Minor concerns:

- p3, l3. The estimates for sepsis incidence do not correspond with the numbers given in the cited reference

- p3, l5. Reference 3 cited by the authors only concerns adults with sepsis, it has no relevance whatsoever to the statement of the sentence

- p3, l11 and p4 l5. the use of the word invalids is inappropriate here (see also above)

- p4, l12. typo, this should probably say SAOD? please check all abbreviations for correctness

- p4, l20. I don't think the word multicentre applies here, please remove. The data are from two sites in the same city and affiliated with the same university.

- p4, l20. This is a retrospective study, patients cannot be recruited

- p6, l1. Supplementary Table 2 does not contain the information the authors refer to

- p6, l1-8. This is another example of poor language. The authors didn't develop a score

- p6, l1-8. I suggest the authors provide this information as a cross table (i.e. for each measure (e.g. SBP, row) they should indicate whether it was used in the respective score (columns pSOFA - SIRS) or not and add the number of missing values)

- p6, l21. Can the authors please add references to decision curve analysis and clinical impact curve

- p6, l22 - p7, l2. Can the authors please clarify what they exactly aimed to do with subgroup analysis, given they had done a multivariate regression model.

- p7, l3. Please explain how multiple imputation was done. Also show the rationale for doing so. Most organ dysfunction scores were developed by assuming that missing values indicate that the value was normal (this does obviously does not apply to systematically missing data as the authors had in their database for pupillary dilation).

- p7, l9. this should be suspected infection

- p7, l16. Why is this Suppl. Fig 3? What happened to Suppl. Fig 1 & 2, they are not mentionned.

- p8, l6-10. This information is given in supplementary figure 3. please add reference and shorten the text accordingly.

- p8, l11-15. It would be better to show a graph with mortality by each increase of 1 point of the different scoring systems as a supplementary figure. Then this section can probably be shortened as well.

- p8, l15-17. The authors presumably report on a very sick population, how is it possible that so many children have a SIRS score <2?

- p8, l20 - p9, l11. No need to formulate "odds ratio of 1.59 equalling an increase of 59% in the relative odds". This is what the odds ratio tells us already. please remove the unnecessary repetition of information.

- p9, l19-20. This is somewhat surprising. Can you specify what the added value of the organ dysfunction scores was in comparison of mortality prediction with the baseline model (AUC of the base line model).

- p9, l22. You mention Brier score here, but it's not discussed in the methods. Please add to methods with proper citation.

- p10, l3-4. The presentation of Brier scores here does not correspond to the way it's shown in Figure 3.

- p10, l5. Typo. this should say 5 and not 8.

- p10, l20-22. Lactate is part of PMODS and PELOD-2, how relevant can subgroup analyses be?

- p13, l1-5. How did you arrive at a lactate threshhold of 2?

p15, l4-7. if you want to suggest lactate should be used with pSOFA, why didn't you look at this in your data? too many missing values?

Tables:

Table 1:

- Please carefully check all number! The percentage for mortality by ethnicity is wrong.

- I suggest to give units for all lab values in the row names and clearly indicate which measure is shown (n (%) versus median (iqr) or (mean (sd) for normally distributed values, see comment above)

- please add the number (%) of children receiving inotropes, ecmo or dialysis to help better assess the disease severity in the population (also relevant regarding mortality)

Figures:

Figure 4: Consider showing score values <2 before >= 2, this would be more logical (0, 1, 2 ...)

Supplementary Figure 4: Please revise the range shown in the DCA plots. it likely dosen't make sense to show risks of mortality > 20-30%. Also consider grouping DCA and CIC in separate panels, or even separate figures to allow the figures to breath (this is in the supplementar material anyway). Please also don't include the graph interpretation in the figure legend.

General concerns with the manuscript presentation:

- The english language is quite poor throughout the mansucript. As PlosOne does not offer copyediting of manuscript I suggest the authors employ a professional service to correct their manuscript (e.g. the authors use the word invalids, probably to indicate children or sick patients on PICU, however I do not think this word is appropriate in the context of the manuscript). There are many other issues with the language, therefore the recommendation to seek professional support.

- please do not repeat all information given in tables or figures in the main text. The authors should reduce such unnecessary repetition (e.g. p7, l10-18; p8, l6-10; or p10, l3-l8)

Reviewer #2: The article is quite interesting, comparing various scores for the prognosis of mortality in the PICU. Strengths include the large sample, which led to a normal or approximately normal distribution of the evaluated scores, and the careful and very informative statistics. Another interesting aspect is that the population is Asian, which fills a gap, as these scores are developed and validated in Western populations.

The article needs to be rewritten for the most part, as it presents language problems throughout the text. For example, the use of expressions such as "invalids" to refer to patients often makes the text difficult to understand.

There are also many typographical errors, such as "morality" instead of "mortality" on pg 14, line 6

There are also many typographical errors, such as "morality" instead of "mortality" on pg 14, line 6

The excessive use of abbreviations also makes the text difficult to follow, and this needs to be reviewed

6. PLOS authors have the option to publish the peer review history of their article (what does this mean?). If published, this will include your full peer review and any attached files.

Reviewer #1: No

Reviewer #2: **Yes: **Orlei Ribeiro de Araujo

---

## [Author Response · Author response to Decision Letter 0]

10 Feb 2024

Reply Letter

Dear Editors and Reviewers,

On behalf of all the contributing authors, I would like to express our sincere appreciation of your letter and the reviewers’ constructive comments concerning our article titled “Validating the performance of organ dysfunction scores in children with infection: A cohort study” (Manuscript No: PONE-D-23-35521). These comments were all valuable and helpful for improving our article. According to the comments of the associate editor and reviewers, we have made extensive modifications to our manuscript and included additional data to make our results more convincing. In this revised version, changes to our manuscript are highlighted within the document in red text. Point-by-point responses to the comments of the academic editor and two reviewers are listed below.

We would like to thank you for allowing us to resubmit a revised copy of the manuscript, and we greatly appreciate your time and consideration.

Thank you, and best regards.

Yours sincerely,

Liping Tan

E-mail: tanlp0825@hotmail.com

---

## [Decision Letter · Decision Letter 1]

17 Apr 2024

PONE-D-23-35521R1Validating the performance of organ dysfunction scores in children with infection: A cohort studyPLOS ONE

Dear Dr. Tan,

Thank you for submitting your manuscript to PLOS ONE. After careful consideration, we feel that it has merit but does not fully meet PLOS ONE’s publication criteria as it currently stands. Therefore, we invite you to submit a revised version of the manuscript that addresses the points raised during the review process.

We look forward to receiving your revised manuscript.

Kind regards,

Dong Wook Jekarl

Academic Editor

PLOS ONE

Journal Requirements:

Reviewers' comments:

Reviewer's Responses to Questions

**Comments to the Author**

1. If the authors have adequately addressed your comments raised in a previous round of review and you feel that this manuscript is now acceptable for publication, you may indicate that here to bypass the “Comments to the Author” section, enter your conflict of interest statement in the “Confidential to Editor” section, and submit your "Accept" recommendation.

Reviewer #3: (No Response)

2. Is the manuscript technically sound, and do the data support the conclusions?

Reviewer #3: Partly

3. Has the statistical analysis been performed appropriately and rigorously? 

Reviewer #3: Yes

4. Have the authors made all data underlying the findings in their manuscript fully available?

Reviewer #3: Yes

5. Is the manuscript presented in an intelligible fashion and written in standard English?

Reviewer #3: No

6. Review Comments to the Author

Reviewer #3: This study is a paper evaluating the performance of the six scoring model using data from a large cohort of pediatric patients. Throughout the paper, there are typing errors and some duplicated sentences. Additionally, there are some points that the authors need to revise.

Despite similar performance between PLEOD2 and SOFA in the conclusion, the reasons why PLEOD2 is not optimal are presented. Please include this information in the abstract (in conclusion section) as well.

Throughout the abstract and the entirety of the manuscript, there is inconsistency in terminologies. Is "age-adapted SOFA" the same as "SOFA" mentioned in the results section? Is this consistent with "paediatric SOFA" as well? Please ensure uniformity and clarity by using one consistent term.

In the abstract, is "severe sepsis" correct, or should it be aligned with the Sepsis-2 criteria or SEVSEPSIS (Table 1) as mentioned in the manuscript? If they differ, please explain the distinction in the introduction. "Sepsis-2" (page 4, line 17) or "sepsis-2" (page 16, line 8)?

Additionally, supplementary Table 1 and 2 incorrectly label "PELOOD2." In the "Lactatemia" section of "PELOOD2," please include the units.(mmol/L?)

Page 3 Ln21 It seems that "age-adapted SOFA" is correct rather than "Adapted SOFA”

Page 5 Ln 19 Systolic Blood Pressure (SBP) should be spelled out before using the abbreviation.

Page 7 Lns12- 21 Please exclude any unnecessary parts from the content of the study cohort. It would be advisable to remove the mention of "Han nationality" from both the table and the main text.

Table 1: It was stated in the method that the worst value on the first day of PICU admission was used for scoring. Is this also the case for the CRP and PCT in Table 1? It would be beneficial to accurately describe this.

Page 9: Lines 11-13 and lines 15-17 contain redundant information. Please review and remove the duplicates.

Page 11 Ln15 It seems that the DCA plot corresponds to Supplementary Figure 7. Please verify and make the necessary correction.

Page 11 The content in lines 19-21 on page 11 overlaps with the content in lines 8-11 on page 10. Editing is required to address this duplication.

Page 12 In this paper, the association between lactate levels and mortality was analyzed. What the analysis results for inflammatory markers such as PCT or CRP are as presented in Table 1? Furthermore, is there any association between these PCT or CRP markers and six scoring models including SOFA? If there is an association, please describe it.

Page 14 Lns 2-5: Please add a reference for lines 2-5 on page 14.

Page 15 Lns 4-7: Please add a reference for lines 4-7 on page 15.

Page 15 Ln13: The sentence at line 13 on page 15 is unnecessary. Please delete it.

7. PLOS authors have the option to publish the peer review history of their article (what does this mean?). If published, this will include your full peer review and any attached files.

Reviewer #3: No

---

## [Author Response · Author response to Decision Letter 1]

16 May 2024

Dear Jekarl,

On behalf of all the contributing authors, I would like to express our sincere appreciation for your letter and the reviewers’ constructive comments concerning our article titled “Validating the performance of organ dysfunction scores in children with infection: A cohort study” (Manuscript No: PONE-D-23-35521). The comments were all valuable and helpful for improving our article. According to the comments of the associate editor and reviewers, we have made extensive modifications to our manuscript to make our results more convincing. In this revised version, changes to our manuscript are highlighted in red text. Our point-by-point responses to the comments of the academic editor and two reviewers are listed below. We employed a professional service, and this manuscript has been edited for proper English language, grammar, punctuation, spelling, and overall style by one or more of the highly qualified native English-speaking editors at AJE.

We would like to thank you for allowing us to resubmit a revised copy of the manuscript, and we greatly appreciate your time and consideration.

Thank you, and best regards.

Sincerely,

Liping Tan

E-mail: tanlp0825@hotmail.com

---

## [Editor Report · Decision Letter 2]

13 Jun 2024

Validating the performance of organ dysfunction scores in children with infection: A cohort study

PONE-D-23-35521R2

Dear Dr. Tan,

We’re pleased to inform you that your manuscript has been judged scientifically suitable for publication and will be formally accepted for publication once it meets all outstanding technical requirements.

Kind regards,

Dong Wook Jekarl

Academic Editor

PLOS ONE
---

## [Editor Report · Acceptance letter]

9 Jul 2024

PONE-D-23-35521R2 

PLOS ONE

Dear Dr. Tan, 

I'm pleased to inform you that your manuscript has been deemed suitable for publication in PLOS ONE. Congratulations! Your manuscript is now being handed over to our production team.

Kind regards, 

on behalf of

Dr. Dong Wook Jekarl 

Academic Editor

PLOS ONE